# Locality and Compositionality in Zero-Shot Learning

**Tristan Sylvain** [*]
Mila, Université de Montréal
Montreal, Canada
`tristan.sylvain@gmail.com`

**Linda Petrini** [*]
University of Amsterdam
Amsterdam, Netherlands
`lindapetrini@gmail.com`

**R Devon Hjelm**
Microsoft Research, Mila
Redmond, USA
`devon.hjelm@microsoft.com`

## Abstract

In this work we study *locality* and *compositionality* in the context of learning representations for Zero Shot Learning (ZSL). In order to well-isolate the importance of these properties in learned representations, we impose the additional constraint that, differently from most recent work in ZSL, no pre-training on different datasets (e.g. ImageNet) is performed. The results of our experiments show how locality, in terms of small parts of the input, and compositionality, i.e. how well can the learned representations be expressed as a function of a smaller vocabulary, are both deeply related to generalization and motivate the focus on more local-aware models in future research directions for representation learning.

## 1 Introduction

A crucial property of a useful model is to *generalize*, that is to perform well on test settings given learning on a training setting. While what is most commonly meant by generalization is being robust to having a limited number of training examples in distributionally-matched settings (Zhang et al., 2016), many tasks are designed to address variations in the data between when a model is trained and when it is evaluated. For instance, some classification tasks address distributional changes in the input: from lacking guarantees of distributional match between train and test (e.g., covariate shift, Shimodaira, 2000) to having fundamental domain differences (e.g., domain adaptation, Crammer et al., 2007; Ben-David et al., 2007). A number of tasks have also been designed specifically to understand models in terms of their ability to generalize to test situations that are poorly represented during training (e.g., Few-Show learning, Li et al., 2006), or even consist of a diverse and entirely novel set of sub-tasks (Zamir et al., 2018). For supervised classification, Zero-Shot Learning (ZSL, Larochelle et al., 2008) is among the most difficult of these tasks, as it requires the model to make useful inferences about (e.g., correctly label) unseen concepts, given parameters learned only from seen training concepts and additional high-level semantic information.

The fundamental question we wish to address in this work is: *What are the principles that contribute to learning good representations for ZSL?* While the most successful ZSL models (Atzmon & Chechik, 2019; Wang et al., 2019) use pretrained features from Imagenet (Krizhevsky et al., 2012; Russakovsky et al., 2015), we wish to understand how these features can emerge given only the data provided from the ZSL task. Specifically, we explore the role of *compositionality* and *locality* (Tokmakov et al., 2018; Stone et al., 2017) as two principles that lead to good generalization. Our study focuses on image representations, so we explore various means of learning representations that are local and compositional for convolutional neural networks (CNNs). We also leverage the structure of CNNs and available annotations from ZSL datasets as a means of interpreting various models in terms of these factors. Overall, our results support the hypothesis that compositionality and locality are crucial principles for training models that generalize well.

---

[*]Equal contribution: order between first authors is arbitrary.

Finally, in order to provide a cleaner framework for understanding the relationship between the above principles and generalization, we re-introduce *Zero-Shot Learning from scratch* (ZFS). In this setting, the model is *not allowed to be pretrained on another dataset*, such as Imagenet, and is evaluated on its ability to perform classification using auxiliary attributes and labels trained *only* using the data available from the training split of the target dataset. We believe that ZFS will provide researchers with a better experimental framework to understand which principles are important for Zero-Shot generalization.

The contributions of our work are as follows:

- We introduce Zero-Shot Learning from scratch (ZFS), an extension to ZSL, which we believe will be an important benchmark for understanding which learning principles lead to better generalization.
- We evaluate several supervised and unsupervised methods on their ability to learn features that generalize in the ZFS setting by training a prototypical network on top of those features (in a similar way to what was done in Snell et al., 2017, with Imagenet features). We then relate this generalization performance with different proxies for locality and compositionality of the given representations, and show that both concepts contribute heavily.
- We introduce a novel version of Deep InfoMax (DIM, Hjelm et al., 2018) which draws local patch representations from other images with the same label as positive samples.
- We introduce a novel visualization technique based on Mutual Information, that allows to investigate local properties of the learned representations.

## 2 Principles that lead to good ZSL performance

Zero-Shot Learning (ZSL, Larochelle et al., 2008) is an important learning framework for understanding a model's capacity to be used in real world scenarios where many relevant test cases (e.g., classes) are not known or are infeasible to sample at training time. An important component of ZSL, particularly in Deep Learning, is learning features directly from raw data (e.g., pixels), that generalize to these test cases. While there are a number of commonly-used strategies for learning generalizable features for various tasks in Deep Learning (Neyshabur et al., 2017), we believe that ZSL in particular requires thinking beyond normal classification by incorporating principles such as compositionality (Boole, 1854) and locality (Fukushima, 1980).

How we formulate, exploit, and analyze these principles to learn models that solve image ZSL tasks will depend on our use of convolutional neural networks (CNNs) as the network architecture for encoding images as well as properties of the data, such as input statistics and available annotations. We will broadly define compositionality and locality, then relate these principles to the tools we have at our disposal from the network architecture and data.

### 2.1 Compositionality

Compositional representations have been a focus in the cognitive science literature (Biederman, 1987; Hoffman & Richards, 1984) with regards to the ability of intelligent agents to generalize to new concepts. Applications are found in computational linguistics (Tian et al., 2016), generative models (Higgins et al., 2017b), and Meta Learning (Alet et al., 2018; Tokmakov et al., 2018), to name a few, with approaches to encourage compositionality varying from introducing penalties (Tokmakov et al., 2018) to using modular architectures (Alet et al., 2018).

In what follows, we will consider a representation to be compositional if it can be expressed as a combination of simpler parts (Andreas, 2019; Montague, 1974). Let $\mathcal{P}$ denote the set of possible parts, $\mathcal{R}$ the representation space and $\mathcal{X}$ the input space. For each $x \in \mathcal{X}$, we assume the existence of a function $D$ mapping $x$ to $\mathcal{P}' \subseteq \mathcal{P}$, the set of its parts. These parts could be local image features (e.g., wings or beaks), or other generative factors (e.g., size, color, etc). Let $g : \mathcal{P} \to \mathcal{R}$ be a function that maps the parts to representations. Formally, $f(x) \in \mathcal{R}$ is compositional if it can be expressed as a combination of the elements of $\{g(p) | p \in D(x)\}$. The combination operator used is commonly a weighted sum (Brendel & Bethge, 2019b), although some works learn more complex combinations (Higgins et al., 2017b). As we consider representations that are implicitly compositional, the above formalism might be approximately true which motivates our later use of the TRE metric.

### 2.2 Locality

Local features have been used extensively in representation learning. CNNs exploit local information by design, and locally-aware architectures have been shown to be useful for non-image dataset, such as

graphs (Kipf & Welling, 2016) and natural language processing (Yu et al., 2018). For supervised image classification, a bag of local features processed independently can do surprisingly well compared to processing the local features together (Brendel & Bethge, 2019a). Attention over local features is commonly used in image captioning (Li et al., 2017), visual question answering (Kim et al., 2018) and fine-grained classification (Sun et al., 2018). Self-attention over local features resulted in large improvements in generative models (Zhang et al., 2018a). Self-supervised methods often exploit local information to learn useful representations: Doersch et al. (2015) proposes to learn representations by predicting the relative location of image patches, Noroozi & Favaro (2016) solves jigsaw puzzles of local patches, and Deep InfoMax (DIM, Hjelm et al., 2018) maximizes the mutual information between local and global features.

For our purposes with image data, we loosely define a local representation as one that has information that is specific to a patch. This helps motivate choices in architecture and learning principles that encourage locality. In this work, we take the most straightforward approach, and use features of CNNs which have receptive fields that are small compared to the size of the full image. This is similar to the motivations in Bachman et al. (2019), where the architecture is carefully designed to ensure that the receptive fields do not grow too large. However, this choice in architecture does not guarantee locality, as CNN representations might hypothetically contain only global information, such as the class or color of the object, despite having a limited receptive field. Therefore, we will evaluate a number of different models on their ability to encode only information specific to those locations. We will discuss relevant evaluation in Sections 4.2 and 4.3.

Note also that compositionality as discussed above and locality are not necessarily independent concepts, nor are they necessarily the same. The set of compositional factors could include local factors, such as parts of an object, but also be more "global" factors, such as general properties of a class (e.g., size, color, shape, etc).

## 2.3 COMPOSITIONALITY AND LOCALITY WITH IMAGE DATA

We focus on three common ZSL dataset that allow us to explore compositionality and locality, namely Animals with Attributes 2 (**AwA2**, Xian et al., 2018), Caltech-UCSD-Birds-200-2011 (**CUB**, Wah et al., 2011), SUN Attribute (**SUN**, Patterson & Hays, 2012). Typical images from these datasets are shown in Fig. 1: CUB is a fine-grained dataset, where the object of interest is small relative to the total image. This is in contrast to AwA2, where subjects have variable size in relation to the total image. SUN is a scenes dataset, meaning that the object of interest is often the whole image.

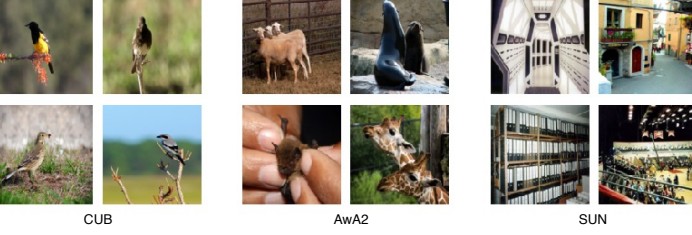

Figure 1: Typical samples show how compositionality and locality are expressed differently in the datasets we consider in this study.

In our evaluation of compositionality, we can leverage different annotations provided by the datasets. All of these datasets provide attributes, which roughly correspond to high-level semantic information composed of a set of underlying factors. For CUB, these attributes describe visual characteristics such as wing colour or shape. For AwA2, these describe both visual and behavioral characteristics, such as the animal's ability to swim or its habitat. For SUN, the attributes are more diverse, ranging from describing the function of the scene, such as playing, to the spatial envelope, for instance man-made. As in Xian et al. (2017), we used $\ell_2$ normalized versions of these attributes as semantic class representations. In addition to attributes, CUB comes with bounding boxes for the whole subject and parts locations for 15 different parts, e.g. beak, tail, belly. These can also be used to assess both locality and compositionality, with the compositional factors being the same as the local ones. The details of each dataset are in the Appendix (Table 1).

# 3 ZERO-SHOT LEARNING FROM SCRATCH

While the original ZSL setting introduced in Larochelle et al. (2008) was agnostic to the exact methodology, more recent image ZSL approaches almost uniformly use features from very large "backbone" neural networks such as InceptionV2 (Szegedy et al., 2016) and ResNet101 (He et al., 2016) pretrained on the Imagenet dataset. In terms of absolute performance, this approach appears to be well-justified, as state-of-the-art results on various ZSL (Yosinski et al., 2014; Sun et al., 2017; Huh et al., 2016; Azizpour et al., 2015) and non-ZSL benchmarks (Li et al., 2019; Zhang et al., 2018b; He & Peng, 2017) all learn on top of similar pretrained backbones.

However, we have many concerns with this approach towards our goal of understanding the principles that contribute to good generalization. First, relative success in transfer learning has been shown to be highly dependent on the precise instantiation of the pretrained backbone encoder (Xian et al., 2018) or the pre-training dataset (Cui et al., 2018). Next, while Imagenet features have been shown to work well for ZSL tasks with similar image datasets, there are no guarantees that a suitable pre-training framework would be available in general ZSL settings. Conversely, it can be hard in practice to meaningfully evaluate a Zero-Shot learner, as performance on specific classes is impacted by their presence in the pre-training dataset (Xian et al., 2017).

Finally, we believe this approach misses the point, in such a way that makes understanding the learning principles that contribute to good generalization difficult. We believe that ZSL should first and foremost be used as a framework for training, understanding, and evaluating models on their ability to reason about new, unseen concepts. Despite the *absolute performance* gains of the methods above that use Imagenet features, the use of backbones hyper-optimized for supervised performance on Imagenet and the Imagenet dataset itself represent *nuisance variables* in a larger effort to understand how to learn generalizable concepts from scratch.

In addition to the ZSL task framework outlined in Larochelle et al. (2008), ZFS simply adds one additional additional requirement: *No model parameters can contain information about (e.g., can be learned from) data outside that from the training split of the target dataset.*

# 4 METHODS

Given our hypothesis on the importance of locality and compositionality, we consider a wide range of representation learning methods trained using the ZFS setting described in Section 3. To showcase the role of these principles, we will introduce a set of proxies for compositionality and locality below. We will also consider auxiliary losses that emphasize locality in the learned representation. Finally, we will introduce a visualization tool that can help identify which local representations are assigned higher importance by each method.

## 4.1 GENERAL APPROACH

In this work, we train convolutional image encoders (CNN) using either supervised or unsupervised learning, then use prototypical networks to perform ZSL transfer on these fixed representations. Prototypical networks are chosen as they require minimal parameters or hyper-parameters tuning, are well-studied (Huang et al., 2019; Finn et al., 2017), and performance is very close to the state of the art for Imagenet-pretrained benchmarks. Our setup is representative of the current state of ZSL models, most of which (Akata et al., 2015; Changpinyo et al., 2016; Kodirov et al., 2017; Zhang et al., 2017; Sung et al., 2018) rely on metric learning by applying two steps: (1) learning a suitable embedding function that maps data samples and class attributes to a common subspace, (2) performing nearest neighbor classification at test-time with respect to the embedded class attributes.

For our study, we compare pre-training the image encoder with a diverse, yet representative set of models:

- *Fully supervised*: Fully supervised label classifier (**FC**)
- *Unsupervised / reconstruction based / generative*: Variational auto-encoders(**VAE**, Kingma & Welling, 2013), $\beta$-**VAE** (Higgins et al., 2017a), Adversarial auto-encoders (**AAE**, Makhzani et al., 2015),
- *Local self-supervision and variants*: Augmented Multiscale Deep InfoMax (**AMDIM**, Bachman et al., 2019) and Class Matching DIM (**CMDIM**).

We pick variants of DIM (Hjelm et al., 2018) as opposed to other self-supervision methods (Doersch & Zisserman, 2017; Noroozi & Favaro, 2016) because extensions have achieved state-of-the-art on numerous related tasks (Veličković et al., 2018; Bachman et al., 2019).

We introduce Class-Matching DIM (CMDIM), a novel version of DIM that draws positive samples from other images from the same class. The goal is to learn representations that focus less on the information content of a single input, while extracting information that is discriminative between classes. The hyperparameter $p$ determines the probability of performing intra-class matching, we experiment with $p \in \{1, 0.5, 0.1\}$. A more detailed description is provided in Appendix G.

**Local classification and attribute auxiliary loss**  We encourage the image encoder to extract semantically relevant features at earlier stages of the network by introducing an auxiliary local loss to the local feature vectors of a convolutional layer (whose receptive field covers a small patch of the image). When used, this auxiliary loss is either from an attribute-based classifier (AC) or a label-based classifier (LC) using the attributes or the labels as supervised signal, respectively. A schematic explanation can been seen in Fig. 7 in the appendix.

### 4.2 Parts classification for CUB evaluation

For each of the $15$ parts labelled in the CUB dataset, we use the MTurk worker annotations to construct $15$ boolean map for each local feature. We project these boolean maps through the CNN to generate ground truth variables that indicate whether the given part is present and *visible* at a location specific to the CNN encoder features. We then train a linear probe for each part, without back-propagating through the encoder, and measure the average F1 score across all locations and parts. This gives us a measurement on how well the encoder represents the parts of the image *at the correct locations*. For more details, please refer to Section F in the Appendix.

### 4.3 Measuring mutual information between local features of different images

As a tool for local interpretability, we propose estimating the mutual information (MI) between global features given from one image and local features from a second image. A schematic explanation is provided in Fig. 8 in the Appendix. In order to do this, we rely on MINE (Belghazi et al., 2018) which uses a *statistics network*, $T_\phi$, with parameters $\phi$ to formulate a lower bound to MI, which is effective for high dimensional, continuous random variables. In our case, the statistics network takes two inputs: a global and local feature vector either sampled from the joint, where each comes from the same image, or from the product of marginals, where the global and local features are sampled independently from each other. The statistics network optimizes a lower bound to the MI:

$$\widehat{I}(G_\theta(X); L_\theta(X)) \geq \mathbb{E}_{p(X)}[T_\phi((G_\theta(X), L_\theta(X))] - \log \mathbb{E}_{p(X) \otimes p(X')}[e^{T_\phi((G_\theta(X), L_\theta(X'))}] \quad (1)$$

Where $p(X) = p(X')$ is the data distribution, and $L$ and $G$ random variables corresponding to the local and global feature vectors of the encoder. At optimum, the output of the statistics network, $T_\phi$ provides an estimate for the Pointwise Mutual Information (PMI), defined as $\log \frac{p(G,L)}{p(G)p(L)} = \log \frac{p(L|G)}{p(L)}$, which roughly gives us a measure of how similar the global and local representations are in terms of information content. Normally, we could try to estimate the marginal term $p(L)$ to get an estimate for the conditional density, but we will normalize our score across the local patches of the target image to get a *relative* score of the relatedness of each local feature to a given global feature. This analysis is similar to that done in Bachman et al. (2019), which looks at different augmentations of the same image using the same MI estimator used to train the encoder.

### 4.4 TRE evaluation

We focus on a metric of compositionality introduced in Andreas (2019), the Tree Reconstruction Error (TRE), as a proxy for compositionality. We will write $\text{TRE}(\mathcal{X}, \mathbf{a})$ for the TRE computed on a dataset $\mathcal{X}$ over the set of primitives $\mathbf{a}$. For details on its definition, see Section H in the Appendix.

As we mostly care about the compositionality with respect to attributes in the context of zero-shot learning and some representations are inherently more decomposable than others (such as VAEs due to the gaussian prior), we consider instead the ratio of the TRE computed with respect to attributes and the TRE computed with respect to uninformative variables (random assignment). We define the TRE ratio as: $\frac{\text{TRE}(\mathcal{X}, \hat{\mathbf{a}})}{\text{TRE}(\mathcal{X}, \tilde{\mathbf{a}})}$, where $\tilde{\mathbf{a}}$ is a random binary matrix (random cluster assignment), and $\hat{\mathbf{a}}$ are the actual visual primitives, attributes in our case.

## 4.5 EXPERIMENTAL SETUP

**Image encoders** We considered both an encoder derived from the DCGAN architecture (Radford et al., 2015) and similar in capacity to those used in early Few Shot Learning models such as MAML (Finn et al., 2017). We also consider a larger AlexNet (Krizhevsky et al., 2012) based architecture to gain insight on the impact of the encoder backbone. It is important to note that overall the encoders we use are significantly smaller than the "standard" backbones common in state-of-the-art Imagenet-pretrained ZSL methods (note that similarly to most recent ZSL methods, the encoder is fixed after pre-training, and used as a feature extractor). We believe restricting the encoder's capacity decreases the performance, but does not hinder our ability to extract understanding of what methods work from our experiments. A detailed description of the architectures can be found in the Appendix (Tables 2 and 3).

**Evaluation Protocol** We used the ZSL splits constructed in Xian et al. (2017), as they are the most commonly used in the literature. All models are evaluated on Top-1 accuracy. We pretrain the encoder using each of the previously mentioned methods (strictly on the considered dataset, as per the ZSF requirement). We then train a Prototypical Network on top of the (fixed) learned representation. All the implementation details are available in the Appendix, in Section B.

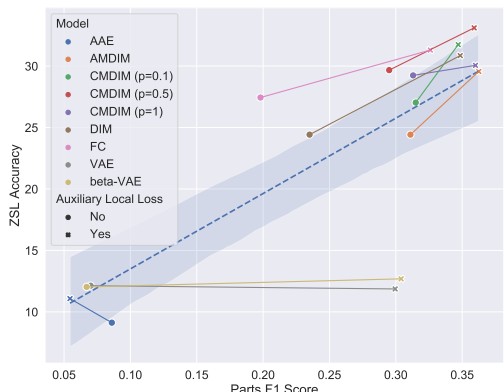

Figure 2: Parts F1 score for all models on CUB with a DCGAN-based encoder plotted against ZSL accuracy. There is a clear relationship between the two: encoders that have a good understanding of local information (as measured by the parts F1 score) perform better in zero-shot learning. The addition of a local loss increases parts F1 score for all models. This improves generalization for all models except those trained with a reconstruction objective.

## 5 RESULTS AND DISCUSSION

In this section, we describe in more detail our experiments and analyze the results. The full numerical results and plots for all considered models can be found in the Appendix.

### 5.1 DOES LOCALITY HELP ZSL, AND CAN LOCAL REPRESENTATIONS BE LEARNED?

**Representations that predict parts at the correct location tend to perform better at ZSL.** We hypothesize that if the encoder represents information that is descriptive of parts, it should also be able to generalize better. To test this, we compare ZSL performance to the part classification F1 score described in 4.2. In Fig. 2, the average F1 scores across the 15 classifiers is plotted against the ZSL accuracy for each model. The two measures are clearly correlated (Person's correlation of 0.73). This relationship doesn't hold for reconstruction-based methods such as VAEs, which could be due to these models needing to represent information related to all pixels, including the background, in order to reconstruct well.

**Encouraging local representations to contain similar information improves ZSL performance** For variants of Deep InfoMax (DIM, Hjelm et al., 2018), AMDIM and CMDIM, the local representations are encouraged to be similar to a global representation through the mutual information maximization objective. While locally specified, this is somewhat contrary to our definition of locality in 2.2. Among the models we tested, these variants generally perform very well, with CMDIM performing the best overall.

This indicates that, while important, locality by itself is not sufficient to learn good ZSL representations. The local representations must also share information, e.g., through a global representation or the class. We hypothesize that such constraints help the encoder learn information that is present locally, but relevant to discriminating the class or important high-level semantic meaning. The above observation also holds for the local losses (AC and LC introduced in 4.1). These losses both encourage the model to

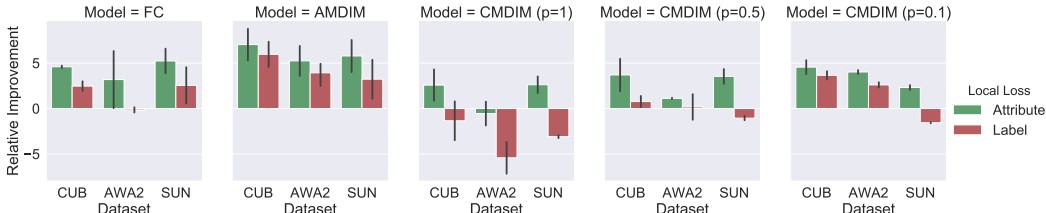

Figure 3: Relative improvement in terms of ZSL accuracy with respect to models trained without the auxiliary loss. Attribute information results in a bigger improvement. Surprisingly, for certain models label information results in a decrease in generalization performance.

rely on local information (local features must capture important semantic information), and for these representations to share information (by having high mutual information with either the attributes or labels).

**Adding local losses helps supervised and self-supervised models.** We investigate in more detail the effect of encouraging the model to take into account different types of local information. As can be seen in Fig. 2, the addition of a local loss improves both ZSL and parts score for all models except the generative ones (VAE, AAE). Interestingly, for these models the parts score also increases, indicating more *locality*, but this does not translate to better ZSL performance.

To better investigate why local losses improve generalization for supervised and self-supervised models, in Fig. 3 we show the relative improvement of each type of local classifier over the performance of the encoder only trained with its global loss. We can see how for the supervised model, the attribute based auxiliary loss has a much bigger impact, which indicates that label information is already exploited by the model, while attributes actually provide more information. For AMDIM, both losses seem to have a consistent positive effect, possibly because the model is unsupervised and hence any label-related information is useful. For CMDIM, the LC auxiliary loss actually hurts performance. This is likely due to the fact that both CMDIM and the LC loss focus on discriminating classes at the local level, and that the LC objective is inherently less effective than the CMDIM formulation for this task (in terms of downstream ZSL performance). As a result, forcing the model to account for both terms lowers downstream performance. DIM and AMDIM discriminate instances and not classes so adding class and attribute information in the form of the AC and LC losses helps performance. CMDIM is already exposed to class information, so only gains from being exposed to the (more informative) attribute information in the form of AC.

## 5.2 How do different models encode information locally?

In Fig. 4 we apply the PMI-based visualization technique introduced in Section 4.3 to pairs of images from the CUB dataset. In this case, we are examining the global representation extracted by each model for the top left image (the Pacific Loon) and comparing it with local features from images of various classes. By noticing which patches each model pays attention to (have higher mutual information), we can infer how information is coded locally. The main take-aways are the following:

- **Supervised models.** The fully supervised model in the first row seems to be able to focus on relevant semantic details, such as the tail of the Horned Grebe and the head of the Back Tern.
- **Unsupervised models.** In the second and third row we can see how models based on a reconstruction loss seem to fail at highlighting semantic information: for images with patterns and colours similar to the Pacific Loon, such as the other Pacific Loon or the White breasted Kingfisher, PMI is high across all local features, while scores are very low for the Tree Swallow with the uniform green background. This could possibly indicate that these models focus more on pixel statistics necessary for reconstruction, and are unable to extract as much semantic information.
- **Self-Supervised models.** AMDIM manages to recover some semantic structure, e.g. for the other Pacific Loon or for the Rusty Blackbird, but fails in some other cases. CMDIM on the other hand, especially for high matching probability $p$, produces heatmaps that are very similar to those of the supervised models, hence managing to recover what is discriminative between classes.

**Models that encode semantic information well perform well on ZSL generalization. Models that reconstruct pixels well perform worse.** To confirm our intuitions on how some families of models focus more on semantic information, while others are more sensitive to pixel statistics, we rely again on the parts binary maps described in 4.2. For each pair of images, we compute the ratio between the score assigned to patches containing *any* part (i.e., a logical OR computed across all binary maps) and

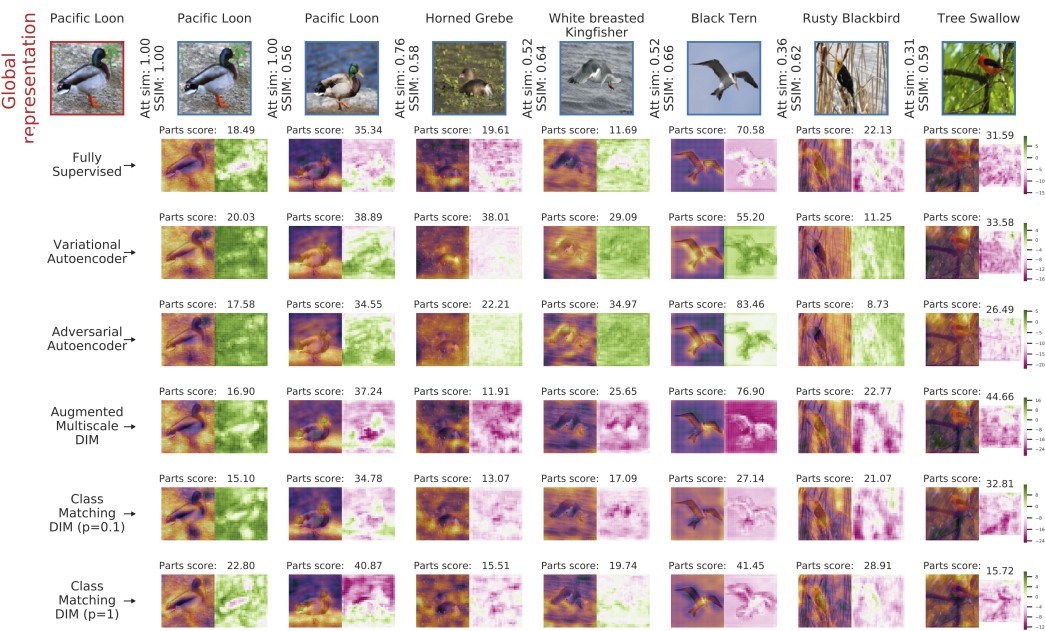

Figure 4: Mutual Information heatmaps allow to understand which local patches contributed the most to the final representation. For each heatmap we plot the absolute values in the rightmost plot and the superposition of the heatmap and the original image on the left, to increase interpretability. Yellow corresponds to higher scores.

the overall score of all features. We refer to this measure as the *Parts ratio*. This ratio is sensitive to both how highly the model scores the relevant parts and to whether the rest of the features are assigned a lower score. We then compute two different types of similarity between the considered images: a semantic one, defined as the cosine distance between the attributes associated to the images' classes, and an pixel-wise one, by measuring the Structural Similarity index (SSIM) between the images. We then measure correlation between the Parts ratio and these measures of similarity. Our interpretation is the following:

- **Positive correlation with attribute similarity:** If two images are semantically similar, the part ratio should be higher if the model manages to extract the common semantically relevant patches (and we know that they are the parts for CUB).
- **Negative correlation with SSIM score:** If a model is too sensitive to pixel similarity, the parts score will be higher for images that are very different, where the only thing in common (pixel-wise) is to depict a bird, while for very similar images the model will just assign a high score to all local features.

We find that for VAEs, the ratio and the attribute similarity are not correlated, but the ratio and the SSIM scores correlate negatively. The effect is reversed for Supervised and Self-Supervised models. This confirms our intuition that VAEs and reconstruction models are not well suited to learn representations that generalize in our context. More details about this experiment and the correlation coefficients are reported in the Appendix, in Fig. 10.

## 5.3 Do compositional representations perform better for ZSL?

Intuitively, we expect compositionality to be an advantage: if a model has a good understanding of how parts map to representations, it can learn to combine known concepts to describe new classes. The experiments show:

- **Measures of implicit compositionality correlate strongly with ZSL performance.** Fig. 5 shows the relationship between the TRE ratio introduced in 4.4, and ZSL accuracy. The Pearson correlation coefficients between the TRE Ratio and ZSL Accuracy are the following: -0.90 for CUB, -0.60 for AwA2 and -0.30 for SUN.
- **The relation is strongest when the attributes are strongly relevant to the image.** For the AWA2 and CUB, datasets for which the attributes are semantically very meaningful, we observe that there is a direct relationship between TRE ratio and ZSL performance. This relationship degrades for

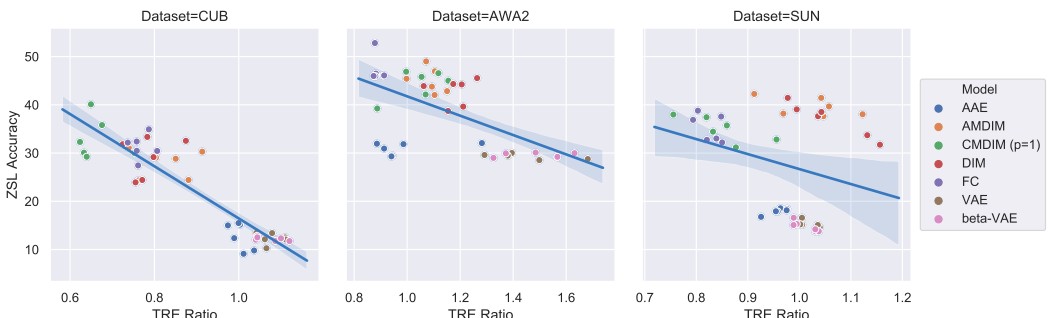

Figure 5: Relationship between TRE ratio and ZSL accuracy for each dataset (lower TRE ratio is better).

SUN, for which the attributes are per-image, and averaged over classes, meaning that they are less likely to map to information actually present in a given image.

We also consider the effect of combining local representations directly (instead of relying on the global output of the model). Given local representations, there are several ways to employ them to perform classification: one option is to create a final representation by averaging the local ones, another option is to classify each patch separately and then average this predictions.

**An explicitly compostional model based on local features helps ZSL.** The results for this comparison are shown in Fig. 6. Averaging representations can be seen as directly enforcing a notion of compositionality: the representation of the whole input is directly built as a weighted sum of the patch representations (that we can imagine being more similar across different data points) and where the weights are uniform. For CUB, and to a lesser extent AwA2, where only few patches encode important information such as beaks, tails, the effect is quite pronounced. There is less of a difference for SUN, where the object is usually the whole scene, meaning all patches are expected to contribute.

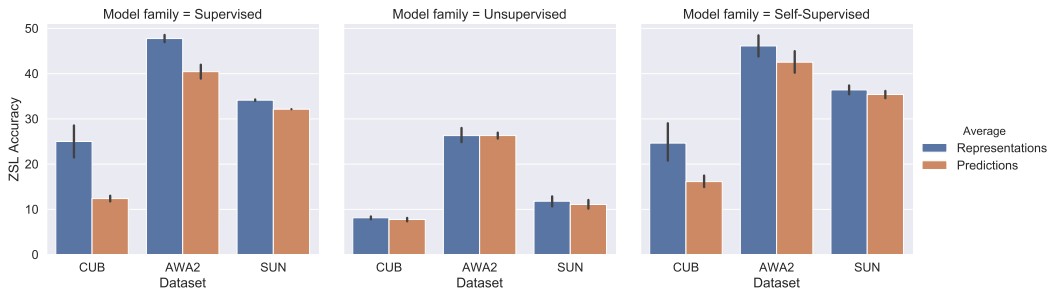

Figure 6: Comparison between averaging representations and averaging predictions. We can see how, for the more local model families, this notion of compositionality is most useful for CUB, where the object of interest is likely only present in few patches.

## 6 CONCLUSION AND FUTURE WORK

Motivated by the need for more realistic evaluation settings for Zero-Shot Learning methods, we proposed a new evaluation framework where training is strictly performed only on the benchmark data, with no pre-training on additional datasets. In the proposed setting, we hypothesize that locality and compositionality are fundamental ingredients for successful zero-shot generalization. We perform a series of tests of the relationship between these two aspects and zero-shot performance of a diverse set of representations. We find that models that encourage both these aspects, either explicitly (through a penalty per instance) or implicitly by construction, tend to perform better at zero-shot learning. We also find that models that focus on reconstruction tasks fail at capturing the semantic information necessary for good generalization, calling into question their applicability as general representation learning methods.

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

## A    EXTRACTING LOCAL AND GLOBAL FEATURES FROM A CNN ENCODER

### A.1    EXPLAINING LOCAL AND GLOBAL FEATURES

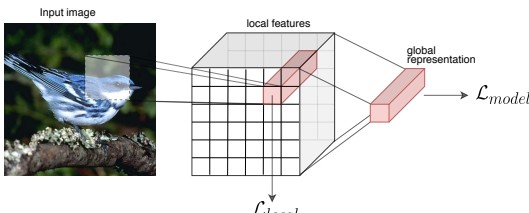

Figure 7: The convolutional encoder takes as input an image and outputs a global representation - used to compute the model loss $\mathcal{L}_{model}$. To encourage locality and compositionality, label or attribute based classification is performed on the activations from early layers($\mathcal{L}_{local}$).

### A.2    EXPLAINING THE ROLE OF LOCAL AND GLOBAL FEATURES FOR MI COMPUTATION

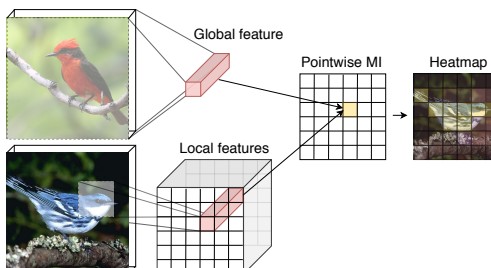

Figure 8: Local and global features are extracted from different images (where local features are activations from an early layer in the CNN encoder), then scored against each other. A high score means the two are considered likely to be extracted from the same image by the model, giving us insight in what information is encoded by the global representation. A heatmap of the scores is used to make the result interpretable.

## B    IMPLEMENTATION DETAILS

All models used in this paper have been implemented in PyTorch, and code will be made publically available. All images were resized to size $128 \times 128$, random crops with aspect ratio $0.875$ were used during training, and center crops with the same ratio were used during test. While most ZSL approaches do not use crops (due to the fact that they used pre-computed features), this experimental setup was shown to be efficient in the field of text to image synthesis (Reed et al., 2016). All models are optimized with Adam with a learning rate of $0.0001$ with a batch size of $64$. The final output of the encoder was chosen to be $1024$ across all models. Local experiments were performed extracting features from the third layer of the network. These features have dimension $27 \times 27 \times 384$ for the AlexNet based encoder and $14 \times 14 \times 256$ for the DCGAN encoder.

## C    MI HEATMAPS COMPARING CMDIM WITH DIFFERENT LOCAL LOSSES

In Fig. 9 we show how local losses affect the type of information CMDIM extracts. We can see how in some cases, e.g. the Nashville Warbler and Rusty Blackbird, the label-based local loss (LC) results in the encoder focusing more on the background and missing out on discriminative features. On the other hand, the label-based (LC) local loss helps the model focus on more localised distinctive patches, especially for the Rusty Blackbird and the Rufous Hummingbird.

## D    PARTS RATIO
### AND RELATIONSHIP TO DIFFERENT MEASURES OF IMAGE SIMILARITY.

In Fig. 10, we plot the Parts ratio against the two different measure of image similarity considered in our experiments and we report the correlation between them across the considered families of models. The correlation was computed over 20.000 pairs of images for each family. While not being a strong correlation, our experiments show how it's a statistically significant one, with associated p-value (expressing the probability of a not-correlated sample resulting in the reported correlation coefficients) of less than $10^{-6}$.

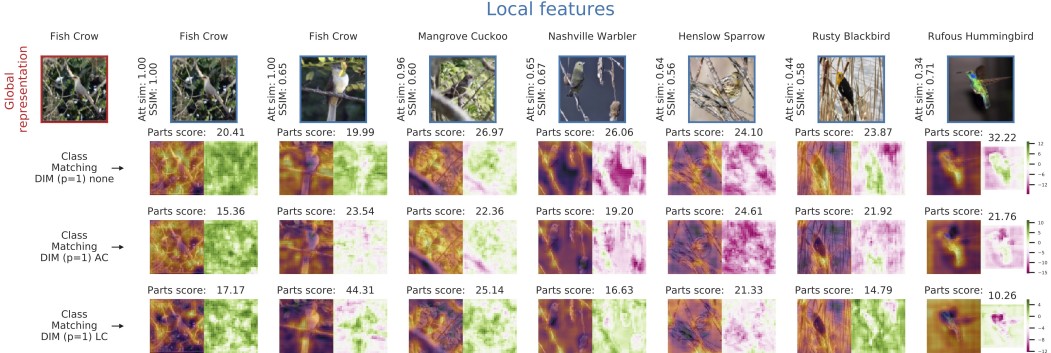

Figure 9: Visualization of locality comparing encoders trained with CMDIM's loss and the proposed local losses. The Figure highlights the impact of local losses on the content extracted by the encoders.

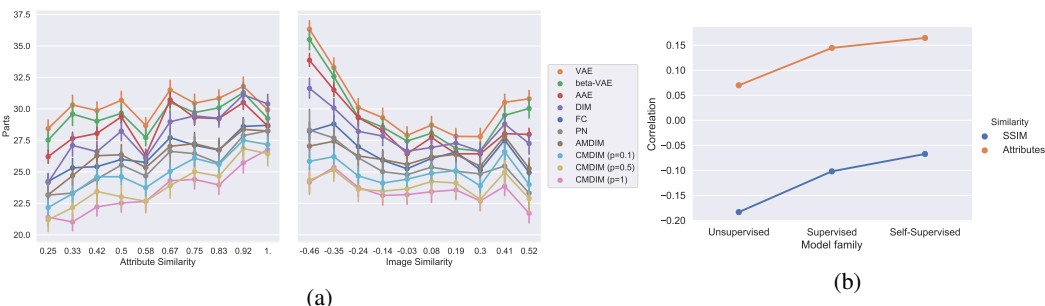

Figure 10: Relationship between the parts score and different measures of similarity. On the left, the parts scores is plotted against the two different measures of similarity. We can see there is a clear trend for all the models: the parts score increases for more semantically similar images, and decreases as the images become more similar pixel-wise. The figure on the right shows Pearson's correlation coefficient between the metrics for different models

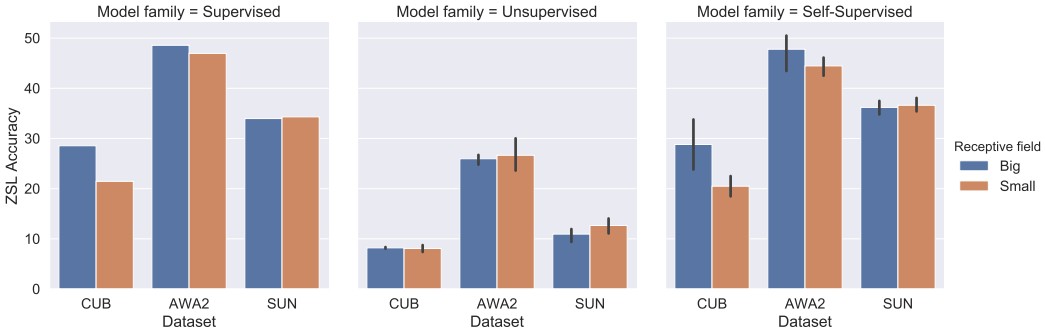

Figure 11: Comparing pre and post-pooling (respectively *Small* and *Big*) features in terms of ZSL accuracy. The effect of a varying receptive field size strongly depends on the model type and on the dataset, highlighting how locality is expressed differently in the datasets.

# E   HOW LOCAL DO THE REPRESENTATIONS NEED TO BE?

A somewhat alternative way to test if local information is relevant for generalization is to explicitly only consider local information. To achieve this, we consider performing classification directly on features whose receptive field doesn't cover the entire image. In our experiments we consider the features extracted from the AlexNet based encoder right before the flattening and final linear layers. To see the effect of varying the receptive field, we perform the same experiment and pre and post last

pooling layer, going from a receptive field of 65 pixels (referred to as *small* in the plots) to 85 pixels (*big*), out of 128 in the original input. The way these local prediction are combined is described in the following section. The results are summarized in Fig. 11. We can see how the dataset seems to make quite a difference for class-matching DIM that benefits from pooling for the datasets where usually the object is in a small part of input, while for SUN, where the whole image tends to be relevant as the images depict scenes, pooling either does not affect or has a negative effect on performance. For reconstruction based models on the other side we see a different trend, where not performing pooling consistently results in better performance across all datasets.

## F    BIRD PARTS LOCATION MAPS

The parts annotations provided with the CUB dataset give us the ability to explicitly quantify whether the encoder is learning to extract meaningful local information. To evaluate this, we train a classifier for each part, that takes as input local features extracted from a specific layer of the CNN encoder and outputs the probability of that part being present within the receptive field of the local feature. To construct a ground truth for this evaluation, we pre-process the parts clicks annotations as follows: the datasets provides, for each input and part, a list of multiple parts location as perceived by multiple MTurk workers. Each annotation is provided as $(x,y)$ coordinates of the center of the part and a boolean feature *visible* indicating whether the part is hidden in the considered input. The ground truth for the classifiers is obtained by converting each part annotation into a boolean semantic map, where a truth value is assigned to a square of side 10 pixels centered in all the locations provided by different MTurk users for each part when visible. This process is repeated separately for all the 15 parts. The obtained boolean masks are then processed through a CNN to project them to the size compatible with the extracted features, so that the classifier's loss can be computed. Importantly, this loss is never backpropagated through the encoder, as these classifiers are meant to only evaluate whether the considered local features are predictive of the parts.

## G    CLASS MATCHING DIM

Deep InfoMax (Hjelm et al., 2018) is a self-supervised representation learning algorithm that is trained by maximizing Mutual Information (MI) between local and global features extracted from the network. More specifically, DIM's objective is a lower bound to MI based on the Donsker-Varadhan representation of the KL divergence that computes two expectations: one over the joint distribution of local and global features, and one over the product of the marginals. In the original DIM setting, samples from the joint distribution (positive samples) are defined as local-global pairs extracted from the same input, while for the product of marginals (negative samples) local and global features are extracted from different inputs.

Class Matching DIM performs a similar operation, but samples from the joint distribution are defined to be pairs of local-global features extracted from *different inputs* belonging to the *same class*, and negative samples are pairs where features are extracted from inputs of different classes. Moreover, we add a hyper-parameter $p$ that allows to control the interplay between DIM and CMDIM, so that positive samples are extracted from inputs of the same class with probability $p$ and from the same input otherwise. Intuitively, this would push the encoder to extract features relevant to a specific input while identifying what features are shared across a single class.

## H    DEFINITION OF TRE

We introduce the following notations:

- $\mathcal{X}$ is a dataset, split into train $\mathcal{X}_{tr}$ and test $\mathcal{X}_{te}$ sets.
- For $x \in \mathcal{X}$ belonging to a class with binary attributes $a_i$ (for continuous attributes we threshold them beforehand), we define $D(x)$ to be the set of 1-valued attributes (present attributes).
- Each attribute $a_i$ is assigned a learnable vector representation $f_\eta(\mathbf{a}_i) = \eta_i$.
- $\delta(\cdot,\cdot)$ is a distance function, chosen to be cosine similarity as in Andreas (2019).

As in Andreas (2019) we combine individual attribute representations by summation:

$$f_\eta(D(x)) = \sum_{\mathbf{a}_i \in D(x)} f_\eta(\mathbf{a}_i)$$

We can now define:

$$\text{TRE}(x,\mathbf{a};\eta) = \delta\Big(f_\eta(x), f_\eta(D(x))\Big)$$

$$\text{TRE}(\mathcal{X},\mathbf{a};\eta) = \frac{1}{|\mathcal{X}|}\sum_{x\in\mathcal{X}}\text{TRE}(x,\mathbf{a};\eta)$$

We compute $\eta = \text{argmin}_{\eta'}\text{TRE}(\mathcal{X}_{\text{tr}},\mathbf{a};\eta')$ and omit it in what follows by abuse of notations.

## I   ARCHITECTURES AND DATASETS DETAILS

Table 1: Details of the datasets used

| Dataset | #Images | #Attributes | #Classes | #Train classes | #Test classes |
|---------|---------|-------------|----------|----------------|---------------|
| CUB | 11,788 | 312 | 200 | 150 | 50 |
| AWA | 30,475 | 85 | 50 | 40 | 10 |
| SUN | 14,340 | 102 | 717 | 645 | 72 |

Table 2: Basic 128x128 architecture details.

| Layer | Layer type | Layer params | pooling | activation |
|-------|-----------|--------------|---------|-----------|
| 0 | conv | (64, 4, 2, 1), batch norm | - | ReLU |
| 1 | conv | (128, 4, 2, 1), batch norm | - | ReLU |
| 2 | conv | (256, 4, 2, 1), batch norm | - | ReLU |
| 3 | conv | (512, 4, 2, 1), batch norm | - | ReLU |
| 4 | conv | (1024, 4, 2, 1), batch norm | - | ReLU |
| 5 | flatten | - | - | - |
| 6 | linear | (1024), batch norm | - | ReLU |

Table 3: AlexNet 128x128 architecture details.

| Layer | Layer type | Layer params | pooling | activation |
|-------|-----------|--------------|---------|-----------|
| 0 | conv | (96, 3, 1, 1), batch norm | (MaxPool2d, 3, 2) | ReLU |
| 1 | conv | (192, 3, 1, 1), batch norm | (MaxPool2d, 3, 2) | ReLU |
| 2 | conv | (384, 3, 1, 1), batch norm | - | ReLU |
| 3 | conv | (384, 3, 1, 1), batch norm | - | ReLU |
| 4 | conv | (192, 3, 1, 1), batch norm | (MaxPool2d, 3, 2) | ReLU |
| 5 | conv | (192, 3, 1, 1), batch norm | (MaxPool2d, 3, 2) | ReLU |
| 6 | flatten | - | - | - |
| 7 | linear | (4096,), batch norm | - | ReLU |
| 8 | linear | (4096,), batch norm | - | ReLU |

## J   TRE RATIO VALUES

Table 4: TRE ratio values for the CUB dataset

| | Normal | AC | LC | Normal_a | AC_a | LC_a |
|---|--------|-----|-----|----------|------|------|
| Model | | Basic | | | Alex | |
| classifier_full | 0.761 | 0.737 | 0.807 | 0.758 | 0.787 | 0.758 |
| vae | 1.062 | 1.042 | 1.066 | 1.111 | 1.079 | 1.11 |
| vae_beta | 1.04 | 1.044 | 1.044 | 1.12 | 1.087 | 1.1 |
| aae | 1.011 | 0.989 | 1.036 | 1.003 | 0.974 | 0.999 |
| local_dim | 0.771 | 0.875 | 0.798 | 0.755 | 0.783 | 0.725 |
| local_twopass | 0.881 | 0.913 | 0.85 | 0.762 | 0.804 | 0.739 |
| local_twopass_class_matching | 0.639 | 0.633 | 0.752 | 0.676 | 0.649 | 0.624 |

Table 5: TRE ratio values for the AWA2 dataset

| | Normal | AC | LC | Normal_a | AC_a | LC_a |
|---|---|---|---|---|---|---|
| Model | | Basic | | | Alex | |
| classifier_full | 0.88 | 0.873 | 0.913 | 0.882 | 0.878 | 0.909 |
| vae | 1.395 | 1.292 | 1.379 | 1.566 | 1.499 | 1.681 |
| vae_beta | 1.325 | 1.33 | 1.372 | 1.631 | 1.485 | 1.567 |
| aae | 0.886 | 0.941 | 0.986 | 1.282 | 0.945 | 0.913 |
| local_dim | 1.211 | 1.206 | 1.174 | 1.154 | 1.264 | 1.062 |
| local_twopass | 1.103 | 1.072 | 1.093 | 1.151 | 0.998 | 1.105 |
| local_twopass_class_matching | 0.996 | 1.155 | 0.887 | 1.054 | 1.118 | 1.07 |

Table 6: TRE ratio values for the SUN dataset

| | Normal | AC | LC | Normal_a | AC_a | LC_a |
|---|---|---|---|---|---|---|
| Model | | Basic | | | Alex | |
| classifier_full | 0.85 | 0.803 | 0.82 | 0.839 | 0.794 | 0.848 |
| vae | 1.005 | 1.005 | 1.002 | 1.04 | 1.036 | 1.035 |
| vae_beta | 0.989 | 0.995 | 0.989 | 1.038 | 1.031 | 1.031 |
| aae | 0.963 | 0.976 | 0.975 | 0.954 | 0.926 | 0.958 |
| local_dim | 1.132 | 0.977 | 1.043 | 1.156 | 0.995 | 1.036 |
| local_twopass | 1.047 | 0.912 | 0.969 | 1.123 | 1.042 | 1.057 |
| local_twopass_class_matching | 0.832 | 0.755 | 0.877 | 0.859 | 0.82 | 0.956 |

## K  F1 PART AVERAGE SCORES

Table 7: Part average F1 score, CUB dataset, basic encoder.

| Loss | Normal | AC | LC |
|---|---|---|---|
| **Model** | | | |
| FC | 0.198 | 0.368 | 0.284 |
| VAE | 0.07 | 0.334 | 0.265 |
| beta-VAE | 0.067 | 0.353 | 0.255 |
| AAE | 0.086 | 0.085 | 0.024 |
| DIM | 0.235 | 0.393 | 0.304 |
| AMDIM | 0.311 | **0.406** | 0.319 |
| CMDIM (p=1) | 0.313 | **0.406** | 0.314 |
| CMDIM (p=0.5) | 0.295 | 0.397 | 0.321 |
| CMDIM (p=0.1) | 0.315 | 0.382 | 0.312 |
| PN | 0.288 | | |

Table 8: ZSL accuracy, comparing the different local losses.

| Encoder | | alex128x128 | | | basic128x128 | | |
|---|---|---|---|---|---|---|---|
| Loss | | Normal | AC | LC | Normal | AC | LC |
| Model | Dataset | | | | | | |
| FC | | 30.47 | 34.92 | 32.40 | 27.44 | 32.17 | 30.44 |
| VAE | | 12.08 | 13.41 | 12.51 | 12.13 | 13.46 | 10.27 |
| beta-VAE | | 11.75 | 11.77 | 12.33 | 12.03 | 12.85 | 12.52 |
| AAE | | 15.16 | 15.00 | 15.49 | 9.12 | 12.36 | 9.80 |
| DIM | CUB | 23.93 | 33.35 | 31.84 | 24.42 | 32.54 | 29.17 |
| AMDIM | | 24.34 | 29.05 | 31.12 | 24.42 | 30.29 | 28.83 |
| CMDIM (p=1) | | 35.80 | **40.11** | 32.31 | 29.24 | 30.08 | 30.04 |
| CMDIM (p=0.5) | | 35.12 | 37.02 | 35.27 | 29.67 | **35.15** | 31.06 |
| CMDIM (p=0.1) | | 29.83 | 33.60 | 33.02 | 27.03 | 32.35 | 31.14 |
| PN | | 37.59 | - | - | 26.29 | - | - |
| FC | | 46.48 | **52.81** | 46.04 | 45.94 | 45.98 | 46.09 |
| VAE | | 29.17 | 28.54 | 28.76 | 30.02 | 29.60 | 29.48 |
| beta-VAE | | 29.98 | 30.11 | 29.22 | 29.00 | 29.47 | 29.94 |
| AAE | | 32.07 | 29.46 | 30.93 | 31.94 | 29.31 | 31.85 |
| DIM | AWA2 | 38.73 | 45.54 | 43.89 | 39.63 | 44.23 | 44.32 |
| AMDIM | | 42.84 | 45.41 | 46.95 | 42.04 | 49.01 | 43.77 |
| CMDIM (p=1) | | 45.80 | 46.56 | 42.14 | 46.87 | 45.00 | 39.70 |
| CMDIM (p=0.5) | | 46.87 | 48.06 | 48.45 | 46.87 | 47.92 | 45.63 |
| CMDIM (p=0.1) | | 47.29 | 51.51 | 50.17 | 45.71 | **49.51** | 48.01 |
| PN | | 46.53 | - | - | 45.23 | - | - |
| FC | | 33.02 | 36.89 | 37.57 | 32.20 | 38.79 | 32.74 |
| VAE | | 14.61 | 15.08 | 14.33 | 15.14 | 16.58 | 15.22 |
| beta-VAE | | 13.80 | 14.20 | 13.79 | 15.08 | 15.29 | 16.58 |
| AAE | | 17.93 | 16.78 | 17.86 | 18.55 | 18.41 | 18.13 |
| DIM | SUN | 31.73 | 39.06 | 37.64 | 33.69 | 41.44 | 38.52 |
| AMDIM | | 38.04 | 41.44 | 39.67 | 37.64 | 42.26 | 38.19 |
| CMDIM (p=1) | | 35.73 | 37.43 | 32.81 | 34.44 | 37.98 | 31.18 |
| CMDIM (p=0.5) | | 37.43 | 40.15 | 36.62 | 35.39 | 39.74 | 34.10 |
| CMDIM (p=0.1) | | 40.01 | **42.05** | 38.51 | 40.56 | **43.13** | 38.93 |
| PN | | 32.00 | - | - | 29.82 | - | - |

Table 9: ZSL accuracy, comparing the different local models.

| | | Average before | | Average after | |
|---|---|---|---|---|---|
| Model | Dataset | pool | no pool | pool | no pool |
| FC | | 28.55 | 21.45 | 13.02 | 11.75 |
| VAE | | 8.37 | 8.06 | 8.33 | 8.15 |
| beta-VAE | | 8.04 | 8.77 | 7.91 | 8.02 |
| AAE | | 8.20 | 7.37 | 7.28 | 6.84 |
| DIM | CUB | 18.48 | 14.39 | 15.07 | 11.81 |
| AMDIM | | 21.45 | 17.68 | 16.67 | 14.04 |
| CMDIM (p=1) | | **34.11** | 22.16 | 13.51 | 15.17 |
| CMDIM (p=0.5) | | 33.48 | 22.88 | **19.07** | 16.35 |
| CMDIM (p=0.1) | | 26.17 | 19.23 | 18.38 | 15.89 |
| FC | | 48.57 | 46.95 | 38.90 | 41.98 |
| VAE | | 26.36 | 30.04 | 25.47 | 26.37 |
| beta-VAE | | 24.78 | 26.32 | 27.63 | 27.11 |
| AAE | | 26.74 | 23.59 | 26.08 | 25.23 |
| DIM | AWA2 | 35.37 | 34.77 | 33.54 | 33.99 |
| AMDIM | | 41.37 | 41.34 | 40.57 | 37.12 |
| CMDIM (p=1) | | 49.66 | 46.26 | **48.16** | 42.57 |
| CMDIM (p=0.5) | | 49.17 | 45.99 | 45.90 | 43.28 |
| CMDIM (p=0.1) | | **50.95** | 44.26 | 44.90 | 37.79 |
| FC | | 33.97 | 34.31 | 32.13 | 32.15 |
| VAE | | 11.48 | 12.84 | 10.73 | 10.14 |
| beta-VAE | | 11.96 | 14.06 | 11.89 | 13.06 |
| AAE | | 9.38 | 11.07 | 9.31 | 11.32 |
| DIM | SUN | 25.82 | 26.83 | 24.80 | 25.56 |
| AMDIM | | 34.04 | 35.19 | 34.51 | 34.58 |
| CMDIM (p=1) | | 37.02 | 36.75 | 33.70 | 36.53 |
| CMDIM (p=0.5) | | 37.98 | **38.93** | 35.53 | **37.29** |
| CMDIM (p=0.1) | | 35.73 | 35.60 | 34.58 | 36.39 |

# L    FULL PLOTS

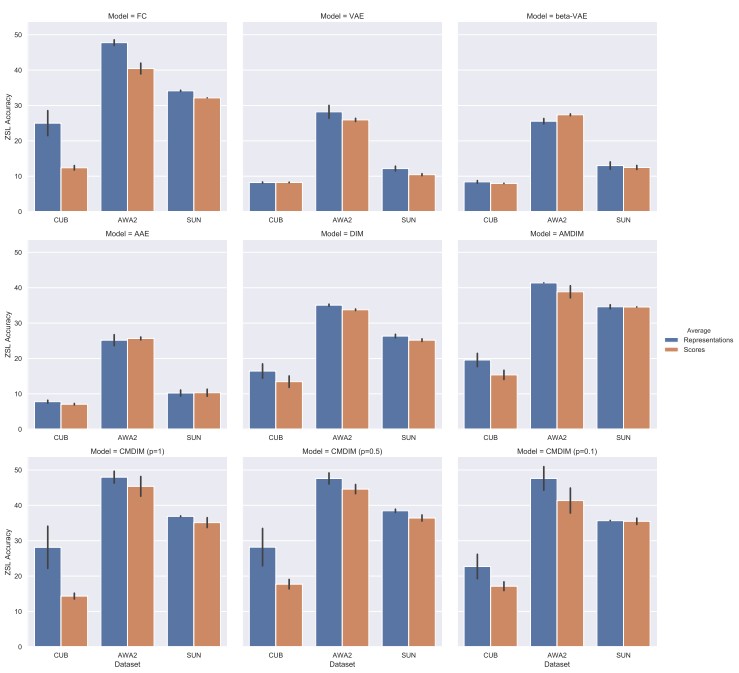

Figure 12: Comparison between averaging representations VS averaging scores for all models.

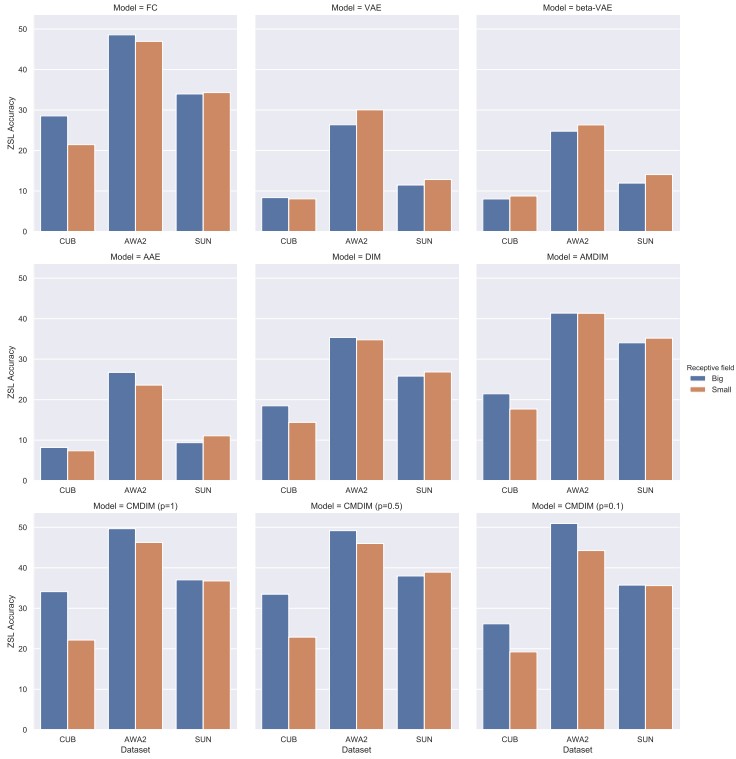

Figure 13: Comparison between small and big receptive field for all models.

