# OpenReview forum: "Locality and Compositionality in Zero-Shot Learning"
_ICLR.cc/2020/Conference — Accept (Poster)_

### Official Review · AnonReviewer1 · 2019-10-22
**Official Blind Review #1**

**Rating:** 6

**Review:**

The paper proposes an evaluation framework for Zero-Shot Learning (ZSL) methods called zero-shot learning from scratch (ZFS) where the model is not allowed to be pretrained on other datasets such as ImageNet. The main motivation of this approach is that it is difficult to understand what is useful for generalization in ZSL since most state-of-the-art methods use features pretrained on large datasets such as ImageNet.
Most state-of-the-art approaches exploit models pretrained on ImageNet. Instead, the paper proposes to randomly initialize model parameters to have a better understanding of what's happening in ZSL.
ZFS adds one constraint: model parameters should not contain information about data outside that from the training split of the target dataset.
Two main criteria are studied to study neural networks:
- compositionality (ability to be expressed as a combination of simpler parts)
- locality (ability to encode only information specific to locations of interest)
To provide a better understanding of their claims, the authors use MTurk annotations to construct boolean map for each local part labelled in the CUB dataset.


Although the paper is experimental, I vote for borderline accept for the following reasons:
- The paper is well-written and the contributions are clear.
- The evaluation metrics to evaluate how models generalize for both criteria are well motivated theoretically (e.g. Tree Reconstruction Error is used to evaluate compositionality), and different types of encoders (e.g. variations of DCGANS) are studied.
- The analysis of models pretrained in different contexts (e.g. supervised classification, reconstruction etc...), with only global or local information, and their impact on generalization is clear (see conclusion for a summary of results). Some conclusions are very intuitive but useful to know (e.g. VAEs and reconstruction models are not well suited to learn representations that generalize in ZFS).


The weaknesses I found in the paper are the following:
- The paper claims that the models used are smaller than the “standard” backbones common in state-of-the-art Imagenet-pretrained ZSL methods. However, few-shot learning method (e.g. ProtoNet) do not retrain the whole model: Protonet freezes most layers and fine-tunes only the last layers to avoid overfitting.
- The paper introduces "Class-Matching Deep Informax (CMDIM)" which draws positive samples from other images from the same class to extract information that discriminative between categories instead of individual samples. However, I did not understand its exact formulation and how it is exactly different from other DIM approaches.


**Experience Assessment:**

I have published one or two papers in this area.

**Review Assessment: Checking Correctness Of Derivations And Theory:**

I assessed the sensibility of the derivations and theory.

**Review Assessment: Checking Correctness Of Experiments:**

I assessed the sensibility of the experiments.

**Review Assessment: Thoroughness In Paper Reading:**

I read the paper at least twice and used my best judgement in assessing the paper.

---

> ### Author Response · Authors · 2019-11-15
> **Author response**
>
> Thank you for your thorough and helpful review.
>
> Regarding the encoder backbone, we do freeze the pre-trained encoder when training Protonet, as you have indicated is common practice in ZSL studies. We apologize that this was not clear, and we clarified this for future readers in the revision by adding information to section 4.5. Our comment on the size of the encoders was only meant to hedge against any readers who might feel the results were too low compared to Imagenet-pretrained results from larger architectures (an example of such an improvement due to the size of the encoder is [1]).
>
> We also clarified CMDIM in our revision, and this can be found in section G of the appendix. To clarify here: CMDIM only differs from DIM / AMDIM in that positive samples are drawn from a mixture of patches from the same image and other images with the same label. As such, CMDIM is not strictly self-supervised, as one reviewer notes, and this can be thought of as training the model to be good at retrieving patches from a query with the same label. We added CMDIM following our observations in Section 5.2 to improve the ability of DIM to recognize similar parts between different images.
>
> Bibliography
> [1] Xian, Yongqin, Bernt Schiele, and Zeynep Akata. "Zero-shot learning-the good, the bad and the ugly." Proceedings of the IEEE Conference on Computer Vision and Pattern Recognition. 2017.

---

### Official Review · AnonReviewer2 · 2019-10-22
**Official Blind Review #2**

**Rating:** 6

**Review:**

UPDATE: My recommendation has been borderline because the discussion of the paper about the nature of locality and compositionality seems to be less in-depth than I would have expected, but if the authors will revise the submission to shift the focus of the paper to more focused on analysis and evaluation and weaken the claims that their model exhibits locality and compositionality, I would lean towards an accept as the empirical evaluation is extensive.

----
Summary: This paper aims to investigate the role of locality and compositionality in zero-shot learning. The authors propose a novel evaluation setup that differs from the original zero-shot learning framework in that the model is not allowed to be pretrained on another dataset. The authors propose several methods for learning and visualizing local representations.

The overall problem the paper tackles is quite ambitious which involve many parts: defining locality and compositionality, defining constraints for enforcing locality and compositionality, evaluating the existence of locality and compositionality, and evaluating model performance. Because of the extensive nature of the research problem, the coverage of each piece of this research problem in this paper could have been more thorough: for example, I would be really excited for a more in-depth analysis on methods for enforcing locality, but the authors consider only one method, which is the auxiliary loss, and this method does not seem to be applicable for datasets that do not have attribute labels. Given the formalism in section 2.1, I would furthermore be excited about a more in-depth analysis on methods for enforcing compositionality, but the authors only investigate compositionality from the perspective of a weighted average. As a result, the claims about whether enforcing locality and compositionality helps with generalization seem weak, as the authors only do not seem to consider different ways of achieving locality and compositionality -- it is not clear whether their method for enforcing locality (with auxiliary losses) or compositionality (by explicitly performing a weighted average) is representative of enforcing locality and compositionality in general. Furthermore, locality seems to be more of a statement about *ignoring* information (discussed below), which the authors do not explore. As a result, my recommendation is borderline. Perhaps the paper could benefit by reframing it with a more specific focus, with more emphasis on the evaluation, which the paper seems to do well. I would recommend that the authors either (1) perform a more extensive analysis of methods for enforcing locality and compositionality or (2) shift the focus of the paper to more focused on analysis and evaluation and weaken the claims that their model exhibits locality and compositionality, as there is not enough evidence to tell whether their method for enforcing locality and compositionality is the best way of doing so.


Strengths:
- The paper is well-motivated and tackles an important problem.
- The empirical evaluation is extensive.
- The framework of not relying on pretrained features is a novel contribution.

Weaknesses:
- There does not seem to be a comparison between a compositional model and a non-compositional model in section 5.3. Would the authors be able to provide such an analysis? Otherwise it is difficult to tell exactly how and to what extent compositionality helps.
- I would have assumed that enforcing locality should actually be a problem of *ignoring* information, rather than *predicting* labeled information. However, the auxiliary loss introduction in section 4.1 only seems to enforce that local features (which by default are local from the CNN) are predictive of attributes/class, rather than enforcing that these features ignore information that is not local to the particular image patch they are modeling. Would the authors be able to comment on this point, as well as provide an empirical analysis of how *unpredictive* the features of their model are on image patches elsewhere?

**Experience Assessment:**

I have read many papers in this area.

**Review Assessment: Checking Correctness Of Derivations And Theory:**

I assessed the sensibility of the derivations and theory.

**Review Assessment: Checking Correctness Of Experiments:**

I assessed the sensibility of the experiments.

**Review Assessment: Thoroughness In Paper Reading:**

I read the paper at least twice and used my best judgement in assessing the paper.

---

> ### Author Response · Authors · 2019-11-15
> **Author response**
>
> We thank the reviewer for the positive comments. In what follows, we will attempt to address their remarks.
>
> Your concerns are completely fair, and I think the issue is that we didn't make it clear enough that the paper is an evaluation paper (and we can correct that). Rather, the methods we introduce are all meant to be different tools we felt would test / evaluate the effect of compositionality and locality. It is true that the paper would have benefited from trying more local / compositional methods, but we think that as establishing a framework for discussing generalization in ZSL, our work has high value, and we hope that future works explore these factors in more depth under ZFS.
>
> Comments on section 5.3. The perspective we have taken in this work is to consider most CNN-based architectures to be compositional in one way or another (given that they arise from combining part representations e.g. feature maps). In this context, it is complicated to introduce a non-compositional model, as it would require a different architecture, hence the use of the TRE proxy and reasoning in terms of degrees of compositionality. Our second experiment in that section (fig. 6) is an attempt to contrast a more compositional model (averaging part representations) with an ensemble (averaging predictions).
>
> Comments on “ignoring” information. We agree that a good criteria for a local representation would be that it ignores information from e.g. other patches. This fits with our proposed definition of a local representation as one that contains information specific to a patch. The reasoning behind the local losses was that as a local feature can only at best be predictive of the class information or attribute information that is present at the given location, the loss would encourage the model to account for this local information. An example would be: a patch corresponding to the beak of a bird cannot predict the tail color, but having an AC loss will encourage it to take into account beak shape, and color. One way to evaluate whether a local representation ignores local information is to estimate the mutual information between different local features corresponding to the same image.  We are working on evaluating this across models and datasets.

---

### Official Review · AnonReviewer3 · 2019-10-24
**Official Blind Review #3**

**Rating:** 8

**Review:**

###  Summary
- The paper tries to understand what learning principles can lead to representations that allow zero-shot learning/generalization.
- It explores the impact of two properties -- locality and compositionality -- on ZSL performance.
- To encourage locality (and also compositionality since they can overlap), the paper uses an auxiliary loss at an earlier layer (when the receptive field of features is small) to predict local attributes.
- For interpreting local features, the paper computes MI between the global feature map of one image and local features from a different image of the same class and visualizes it using a heatmap.
- It uses normalized TRE to measure compositionality (Normalized to take into account that some methods are biased towards learning more compositional representations.)
- It compares supervised, unsupervised, and self-supervised representation learning methods and studies the impact of locality and compositionality on ZSL performance for these methods.

### Decision and reasons
I vote for a weak accept. However, I think it's a good paper and does a lot of things right. I'm willing to increase my score if the authors can convincingly answer my questions.

Positives:
1- The proposed zero-shot learning from scratch setting is a step in the right direction for focusing on uncovering general learning principles.

2- Locality and compositionality are sensible goals for good representations. The paper defines both and explores their importance with clear and novel experiment-designs and metrics. The experiments are also well conducted.

Negatives:

1- The paper makes a lot of observations, but sometimes does not even try to explain some unexpected observations.
2- CMDIM is presented as a self-supervised algorithm but seems to require class labels.

### Supporting arguments for the reasons for the decision.

Positives:

1- I agree with the paper that ZSL setting is more about uncovering learning principles and less about constructing practical systems that can do well in zero-shot settings. In a practical setting, it doesn't make much sense to zero-shot learning anyway. I also agree that current ZSL work is focusing too much on pushing the state-of-the-art using pre-trained imagenet features and missing the actual goal of the problem setting (I wouldn't say we shouldn't do that at all, however. Some learning principles may require a significant amount of data to learn good representations for zero-shot learning, and imagenet pretraining is a good proxy for that). This paper acts as a good reminder that ZSL research should be done keeping in mind the goal of ZSL.

2- The paper first defines locality and compositionality. It then uses interesting and meaningful metrics for empirically testing if these properties correlate with zero-shot performance. I found the experiment designs to be clever (such as computing parts F-1 score using boolean maps, visualizing MI between local and global features of images from the same class, etc).  I think this paper will act as an important reference for motivating future work on learning more modular representations.

Negatives:

1- The paper doesn't try to explain the possible reasons behind some observations. For example, why does locality not correlate with ZSL performance of generative models? Why does LC loss hurt performance for CMDIM?
2- CMDIM draws positive samples from other images of the same class. As a result, it's not a truly unsupervised learning method. The direct comparison of CMDIM to DIM seems unfair.

### Questions

Q1-  Computation of the local classification and attribute auxiliary loss is a bit unclear. How is it assured that a certain attribute is present in the local feature when computing the auxiliary attribute-loss?

Q2- Did the authors try a baseline in which AC is also trained on the global representation in the context of Figure 2? The extra information about the parts may be the reason behind better ZSL performance, and a global AC could improve ZSL performance without improving Parts F1 score (Although a more likely outcome is that a global AC would increase both the parts F1 score and the ZSL accuracy.)

Q3- For VAEs, the local F-1 score does not correlate with ZSL performance. This seems to contradict the idea locality leads to better ZSL performance. Is there an explanation for this? The paper just glances over this by calling this 'interesting.'

Q4- Can CMDIM be considered a self-supervised algorithm? Doesn't it need class labels to draw positive samples from the same class?


### Other minor remarks

- "Formally, f(x) \elem R is compositional if it can be expressed as a combination of the elements of ..."

The definition would be more clear if the authors could mention some reasonable combination operators here (such as weighted average etc).

- "However, this choice in architecture does not guarantee locality, as CNN representations could only contain “global” information, such as the class or color of the object, despite having a limited receptive field. "

I'm not sure what this means. Why are CNN representations restricted to 'only' global information?

### Update after author's response

The author's response fixes some minor clarity issues. I think this work is a good contribution and should be accepted. I've updated my score accordingly.

**Experience Assessment:**

I have published one or two papers in this area.

**Review Assessment: Checking Correctness Of Derivations And Theory:**

N/A

**Review Assessment: Checking Correctness Of Experiments:**

I carefully checked the experiments.

**Review Assessment: Thoroughness In Paper Reading:**

I read the paper thoroughly.

---

> ### Author Response · Authors · 2019-11-15
> **Author response**
>
> We thank the reviewer for their supportive comments and helpful suggestions. In what follows we will attempt to address their concerns, grouped by category.
>
> General observations
> Why does locality not correlate with ZSL performance for generative models?
> We will address this question in our answer to Q3 below.
>
> Why does LC loss hurt CMDIM performance?
> 	We hypothesize this is due to the fact that both CMDIM and the LC loss focus on discriminating classes at the local level, and that the LC objective is inherently less effective than the CMDIM formulation for this task (in terms of downstream ZSL performance). As a result, forcing the model to account for both terms lowers downstream performance. A more detailed explanation has been added to 5.1.
>
>
> Questions.
> Q1. We do not have ground truth values for the local presence or absence of an attribute (this might be feasible on CUB by combining part information with attributes which sometimes map to specific parts such as tail color, but impossible on SUN and AWA2 by construction). As a result, we consider the ground-truth value for each local feature (attribute or class) to be that of its class. This means that some uninformative features are counted in the loss term (e.g. a local feature corresponding to pure background will not be able to predict the class or the attributes) but on average, the loss term encourages local features to capture class/attribute information when possible. More complex setups in the field of attribute detection tend to use attention mechanisms to mitigate the effect of taking into account uninformative local features (such as a feature mapping to the background of the image) [1] but are outside the scope of this work.
>
> Q2.
> We initially ran experiments where we considered versions of AC and LC on the global representation. For AC this is equivalent to multi-task learning combining the model and a ProtoNet. For LC the same applies for a supervised classifier.
> The addition of the global loss changed learning dynamics (e.g. VAEs tended to ignore the reconstruction term) and overall hyperparameter selection resulted in either:
> Weaker (in terms of ZSL accuracy) models focusing on the AC and LC at the expense of their own loss term
> Stronger models mainly ignoring AC and LC.
> The part accuracy overall changed in a similar fashion: either staying close to the original model (ignoring AC/LC) or converging towards that of FC/PN. As this did not provide useful insights into the original model, we dropped this line of investigation. We believe local AC/LC behave differently as local versions of PN and FC do not dominate the performance of other models.
>
>
> Q3.
> This observation is actually more generally true for all models that include a reconstruction objective, such as VAEs but also AAEs, as outlined in figure 2. We hypothesize that this is due to the fact that such models already have a good understanding of local information (essential for reconstructing) which is not really improved by the local objectives. The main issue with those models is that they might be “too local”: focusing on capturing non-essential (from a class-discriminative point of view) local information, such as an image background, at the expense of more important aspects of an image.
> In addition to this, these models score poorly in terms of the measures of compositionality we have introduced. While we have considered locality and compositionality separately, they interact, and a model that is aware of local parts, but unable to combine them effectively will be weaker at ZSL.
>
> Q4. CMDIM is indeed a (weakly) supervised learning algorithm and does draw positive samples from the same class. Overall the goal of introducing this model is to show the improvement one can achieve by changing the nature of positive/negative samples in DIM to learn class-discriminative rather than instance-discriminative features. We have clarified the category name in 4.1 to highlight this (we put it in the same category as DIM/AMDIM due to conceptual similarities). A detailed description of CMDIM has been added to section G of the appendix.
>
> Other remarks.
> “Formally …”. We have updated section 2.1 with examples of common combination operators, referencing more complex ones as found in e.g. [2]
> “However … field.”. We realize that this sentence was not clear and could lead to a misunderstanding. We clarified section 2.2 accordingly.
>
>
> Bibliography
> [1] Liu, Xiao, et al. "Localizing by describing: Attribute-guided attention localization for fine-grained recognition." Thirty-First AAAI Conference on Artificial Intelligence. 2017.
>
> [2] Higgins, Irina, et al. "Scan: Learning hierarchical compositional visual concepts." arXiv preprint arXiv:1707.03389 (2017).

---

### Decision · Program_Chairs · 2019-12-19

**Decision:**

Accept (Poster)

**Comment:**

This paper investigates the role of locality (ability to encode only information specific to locations of interest) and compositionality (ability to be expressed as a combination of simpler parts) in Zero-Shot Learning (ZSL). Main contributions of the paper are (i) compared to previous ZSL frameworks, the proposed approach is that the model is not allowed to be pretrained on another dataset (ii) a thorough evaluation of existing methods.

Following discussions, weaknesses are (i) the proposed method (CMDIM) isn't sufficiently different or interesting compared to existing methods (ii) the paper does not do an in-depth discussion of locality and compositionality. The empirical evaluation being extensive, the accept decision is chosen.